# Factorial Validity of the Pediatric Nurse Parent Partnership Scale—Pediatric Nurses (PNPPS-PN)

**DOI:** 10.3390/healthcare10060991

**Published:** 2022-05-27

**Authors:** Mi-Young Choi, Ju-Yeon Uhm

**Affiliations:** 1Department of Nursing Science, Chungbuk National University, Chungdae-ro 1, Seowon-Gu, Cheongju, Chungbuk 28644, Korea; myb98@chungbuk.ac.kr; 2Department of Nursing, Pukyong National University, Yongso-ro 45, Nam-Gu, Busan 48513, Korea

**Keywords:** factor analysis, statistical, pediatric nursing, nurses, parents, partnership

## Abstract

For the optimal growth and development of high-quality professional nursing care for hospitalized children, a partnership between pediatric nurses and parents within the context of the family is essential. The aim of the present study was to evaluate the factorial validity of the Pediatric Nurse Parent Partnership Scale for hospital pediatric nurses (PNPPS-PN) and to investigate the associations of its components with the Nursing Professional Value Scale and Compassion Competence Scale. This study used survey data from 236 pediatric nurses collected between March and June 2019. The seven-factor structure showed an acceptable to good fit in the confirmatory factor analysis, and the scale correlated as expected with measures of compassion competence and nursing professional values. The Cronbach’s α for the scale was 0.89, indicating good internal consistency reliability. The PNPPS-PN is a psychometrically sound instrument for assessing pediatric nurse-reported partnerships between pediatric nurses and parents in hospital settings.

## 1. Introduction

In a hospital, pediatric nurses and parents are both crucial to providing care to a child. In pediatric nursing, it is the collaborative relationship between pediatric nurses and parents that result in optimal care for the child [1]. During a child’s hospitalization, there are constant interactions between the child, parents, and pediatric nurses. For a pediatric nurse who cares for inpatient children, relationships and interactions with the child’s parents are the basis for providing quality nursing care for hospitalized children [2]. These types of interactions are routine for many nursing situations; nevertheless, they are integral to forming trust-based cooperative relationships between pediatric nurses and parents [1,3]. Nurses who consider their role within the family context and attach an importance to the nurse–parent interaction improve their understanding of the child, and therefore, the care of the child. One of the important functions of the pediatric nurse–parent partnership is recognizing the possibility that the parents are best at identifying the personal needs of their child and describing the child’s unique characteristics [4]. Recently, partnerships between pediatric nurses and parents are viewed as the core tenets for providing quality of nursing care to child patients [5,6,7,8]. In a study to evaluate the effectiveness of the mother–nurse partnership program for mothers with infants with congenital heart disease, the mother–nurse partnership program improved collaboration between pediatric nurses and mothers, and reported positive emotional and cognitive outcomes for mothers of infants undergoing cardiac surgery [7]. In addition, in a study of mothers of infants who had undergone cardiac surgery conducted by a partnership program between critical care nurses, it was found that mothers had a need for information and participation in care related to their child’s condition [9]. Therefore, it is necessary to clearly conceptualize and explore the pediatric nurse–parent partnership as a key component in pediatric nursing. It is also necessary to have a reliable and valid measure of the partnership between pediatric nurses and parents.

Pediatric nurses know the importance of family-centered care (FCC); however, they experience difficulty implementing this approach due to a lack of clarity. Research has been conducted on the NICU context [10] and the path model for the quality of pediatric nursing care [11]. Notably, the partnership between pediatric nurses and parents with hospitalized children is critical to the care and recovery of children [1,5]. Unfortunately, while the need for and importance of the pediatric nurse–parent partnership is well known and emphasized, relatively little research has been conducted to measure this partnership relationship quantitatively [6]. Research has been conducted on measuring FCC [12] and quality of care through the patient’s perspective, or child and parent perceptions of care [13], which are used to measure the quality of child- and family-centered nursing in children’s wards. The partnership between pediatric nurses and parents means that pediatric nurses work together in a way that respects, involves, and empowers parents. The partnership between pediatric nurses and parents is led by child nurses and parents [1,14]. However, few studies have been conducted to develop an instrument that can measure pediatric nurse–parent partnerships, except for the Pediatric Nurse Parent Partnership Scale [PNPPS-PN] [1]. Following the development of this instrument, research has focused on simple correlations between the pediatric nurse and parent partnership and other variables: pediatric nurses’ professionalism and self-efficacy [15], the factors associated with partnership [16], job stress [17], and related variables of the quality of nursing care [18]. Some researchers have determined the effectiveness of a mother–nurse partnership program for attending infants with congenital heart defects [9] and have explored mothers’ needs [7]. However, prior research exploring this partnership between pediatric nurses and parents led to the development of an original scale, which, however, included only an exploratory factor analysis. Given that their construct concept may be different, this study tried to examine the construct validity of the partnership between child nurses and parents, for nurses engaged in pediatric nursing care. In this context, there have not been sufficient studies confirming the construct validity and reliability of the PNPPS-PN.

Gallant et al. [19] indicated that partnerships might be realized through the processes involved in establishing and maintaining relationships. They reported that a partnership could start with an inter-partner agreement allowing both parties to focus their partnership and establish the necessary roles for each; this would involve some power-sharing and negotiation. Young children, who are generally poor at verbal expression and unable to decide on or determine their own health problems, tend to rely on their parents [20]. In particular, they are more likely to have their parents make most of the decisions related to themselves; when a child is hospitalized, the parent, as a caregiver, participates in many aspects of nursing [13,21]. Shields et al. [22] suggested that this could satisfy the physical and emotional needs of the child. Thus, it is necessary to determine the anxiety, stress, and coping styles of a family or parents as well as the hospitalized child, so that the pediatric nurse can provide personalized nursing to meet the needs of the child and the parents [21]. While it was once difficult for the parents and members of the family to stay in pediatric wards, many countries have recently begun permitting a caregiver to stay in the ward with a child [23]. Parental participation means that the parent plays the role of a co-caregiver rather than being excluded from choosing and deciding on nursing for the hospitalized child. This has been recognized as a key principle of pediatric nursing in policies for pediatric health in several countries, including Canada, Sweden, and Australia [21,24,25]. In other words, a parent serves as a partner who expresses the child’s preferences and expectations, in contrast to a passive beneficiary or agent, in providing quality nursing care to the child [1]. Parents, as professionals with expertise and knowledge about their children, provide important information related to their children to healthcare providers such as nurses and doctors [26]. 

It is important for pediatric nurses to form partnerships with parents for providing optimal nursing care to hospitalized children. There is insufficient evidence to examine whether pediatric nurses’ perceived parental partnerships are associated with pediatric nursing performance and outcomes in children. Therefore, studies are needed to confirm the construct factors of partnership between pediatric nurses and parents.

## 2. Objective

Research on the aspects of the pediatric nurse–parent partnership that are essential to nursing hospitalized children and the development of relevant instruments is still limited and ambiguous [1]. Therefore, there is a need for a reliable and valid instrument that can assess pediatric nurse–parent partnerships, as this relationship is known to be a preceding requirement for optimal care. The pediatric nurse–parent partnership scale for pediatric nurses [PNPPS-PN] was first published in 2013. It is necessary to re-validate the factor structure of the scale. The aim of the present methodological study was to re-examine the factorial structure of the Pediatric Nurse Parent Partnership Scale (PNPPS-PN) using confirmatory factor analysis (CFA) and provide evidence of the scales’ validity.

## 3. Methods

### 3.1. Study Design and Participants

This validation study had a cross-sectional and descriptive design. To evaluate the factorial validity of the PNPPS-PN for pediatric nurses, the scale was administered to pediatric nurses at four hospitals in South Korea. Four tertiary hospitals with pediatric nurses caring for a variety of hospitalized children were selected in Seoul, Busan, or Chungbuk Province. Pediatric nurses included nurses working in the general pediatric ward, pediatric cancer ward, neonatal intensive care ward, or pediatric intensive care ward that required constant interaction with the parent of the child. Only nurses with more than 12 months of hospital experience were included. The reason for limiting the subjects to nurses who have worked for more than 12 months was based on the findings that there was a difference in partnerships between nurses working less than 12 months and those working more than 12 months in pediatric wards [27]. Data were collected from March to July 2019. After seeking cooperation for the survey from the nursing department of the hospital, those who voluntarily agreed in writing to participate were selected. We collected data from 240 participants. Of these, four returned scales with unanswered items and these were discarded. Data from the remaining 236 were analyzed. The minimum sample size for confirmatory factor analysis is 200 or more [28]. In addition, to obtain a stable factor structure, a sample size of 170 or more is required, since it is needed at 5 times the number of items (34 items) [29]. Thus, the number of participants in this study was acceptable. The questionnaire took approximately 15–20 min to complete. Participants received gifts (beverage gift cards) as a token of appreciation for their participation after completing the questionnaire.

### 3.2. Instruments

#### 3.2.1. General Characteristics

The general characteristics questionnaire asked for information on participants’ age, marital status, education, religion, current work unit, total years of career experience, and number of years working in pediatric units.

#### 3.2.2. Pediatric Nurse Parent Partnership Scale (PNPPS-PN)

The PNPPS-PN was adopted by Choi and Bang [1]. This scale was developed as a self-reported questionnaire based on the results of a concept analysis using the hybrid model. The scale was originally developed for use with pediatric nurses and parents of hospitalized children. In this study, the partnership between pediatric nurses and parents refers to a cooperative relationship in which child nurses and the patient’s parents provide nursing care together, to provide optimal nursing care for the child, as perceived by pediatric nurses. Thus, factorial validity was confirmed for pediatric nurses. The PNPPS-PN comprises 34 items, including seven subscales: Reciprocity (nine items), Professional knowledge and skill (seven items), Sensitivity (six items), Collaboration (three items), Communication (four items), Shared information (three items), and Cautiousness (two items). Each item is scored on a 5-point Likert scale ranging from 1 (strongly disagree) to 5 (strongly agree), with higher scores indicating that pediatric nurses positively perceived the partnership between themselves and parents. At the time of development, the Cronbach’s alpha for the scale was 0.96, and the Cronbach’s alpha of the subscales ranged from 0.66 to 0.93. 

#### 3.2.3. Nursing Professional Values Scale (NPVS)

Nursing professional values were measured using the Korean version of the Nursing Professional Values Scale (NPVS) devised by Yeun et al. [30]. The scale comprises 29 items grouped under five subscales: Self-concept of the Profession (nine items), Social Awareness (eight items), Professionalism of Nursing (five items), the Roles of Nursing Service (four items), and Originality of Nursing (three items). Each item is scored using a 5-point Likert scale ranging from 1 (absolutely not) to 5 (always), with higher scores indicating a greater degree of nursing professional values. 

#### 3.2.4. Compassion Competence Scale (CCS)

Compassion competence was measured using the Korean version of the Compassion Competence Scale (CCS) devised by Lee and Seomun [31]. The scale comprises three subscales with a total of 17 items: Communication (eight items), Sensitivity (five items), and Insight (four items). Each item is scored using a 5-point Likert scale ranging from 1 (absolutely not) to 5 (always), with higher scores indicating a greater degree of compassion competence. 

### 3.3. Data Analysis

The data were analyzed using SPSS (Statistical Package for the Social Sciences) for windows version 27.0 and AMOS (Analysis of MOment Structure) version 27.0 (IBM Corp., Armonk, NY, USA). Participant characteristics were analyzed using frequency, percentage, mean, and standard deviation. The construct validity of the scale was evaluated using CFA. Factor analysis for scale validation is based on prior theories, and it is desirable to perform CFA with the objective of determining the structure of the factors [32]. CFA of the data was performed to test the factor structure proposed by Choi and Bang [1]. In the originally developed scale, 7 factor structures were confirmed through hybrid concept analysis: Reciprocity, Professional knowledge and skill, sensitivity, collaboration, communication, shared information, and cautiousness. In addition, criterion validity was evaluated by correlating the PNPPS with measures of nursing values and compassion competence.

Based on the values of several model fit indices, the Goodness-of-fit indices for the data were assessed and compared: Chi-square (χ2) value, Chi-square minimum/degree of freedom (CMIN/DF), Comparative Fit Index (CFI), root mean square error of approximation (RMSEA), Goodness-of-Fit Index (GFI), Normed fit Index (NFI), Tucker–Lewis index (TLI), and Akaike’s information criterion (AIC). A CMIN/DF of 2 or less is good. The CFI, GFI, NFI, and TLI are generally considered good if they are 0.9 or higher. In general, the RMSEA of 0.08 to 0.05 is good. The AIC is used to compare two or more models, and the lower the AIC, the better the fit. Correlations between factors were analyzed using Pearson’s correlation. Convergent and discriminant validity were assessed using the average variance extracted value (AVE) and construct reliability (CR) [33]. Further, between models were compared using the chi-squared difference test. Cronbach’s alpha coefficients were used to analyze the internal consistency reliability of the PNPPS-PN. To evaluate convergent validity, the PNPPS-PN was correlated with the CCS and NPVS.

### 3.4. Ethical Consideration

Ethical approval was obtained from the Institutional Review Board of the researcher’s university (CBNU-201812-SB-751-01). Informed consent was obtained from all the participants. Participants were informed of the purpose of this study before consenting to participate. The survey took about 10 min to complete. After the survey was completed, each respondent received approximately $5 as a token of appreciation.

## 4. Results

### 4.1. General Characteristics

All participants were women, and the mean age was 29.29 ± 5.49 years (range 23–49). The average length of total career was 6.53 ± 5.47 years (range 1–26) and the average length of career in the pediatric unit was 5.90 ± 4.62 years (range 1–25). One hundred participants (42.2%) worked in newborn and neonatal intensive care units or pediatric intensive care units, and 76 participants (32.1%) worked in pediatric wards (Table 1).

### 4.2. Construct Validity

#### Confirmatory Factor Analysis

The results of 7-factor models are summarized in Table 2. The model was compared using confirmatory factor analysis. Model 1 confirmed the CFA results with the factor structure and number of items at the time of scale development. Model 2 is the result of the CFA by checking the CFA result of Model 1 and removing 5 items with a standardized regression weights (SMC) value of 0.5 or less. Model 3 is the result of the CFA by checking the CFA results of Model 2 and removing 10 items with an SMC. When comparing fit across the models, Model 3 (the 7-factor structure, 19 items) was superior to Model 1 (original model: the 7-factor structure, 34 items) and Model 2 (the 7-factor structure, 29 items). Although all models showed significant chi-square values (*p* < 0.001), the AIC was lower in Model 3 (427.52) compared with Model 1 (1358.55) and Model 2 (1034.42). The conceptual framework of the original PNPPS-PN and the items were reviewed to categorize each factor as follows: Reciprocity, Professional knowledge and skill, Sensitivity, Collaboration, Communication, Shared information, and Cautiousness. The model fit of the final PNPPS-PN was tested with CFA. The fit of the model was found to be acceptable: χ^2^ = 309.52, CMIN/DF = 2.36, CFI = 0.91, RMSEA = 0.078, GFI = 0.88, NFI = 0.85, TLI = 0.88, and AIC = 427.52. In general, it showed good model fit. Since the CMIN/DF value was less than 3, the overall fit could be satisfied [34]. The items are responded to using a 5-point Likert scale ranging from 1 (strongly disagree) to 5 (strongly agree). Scores can range from 19 to 95, with higher total scores indicating that the pediatric nurses perceive that they have high-level nurse–parent partnerships. The average score of 19 items was 4.03 ± 0.37, and the range was 2.95–5.00.

### 4.3. Convergent and Discriminant Validity

Convergent and discriminant validity were assessed using the AVE value and CR. The AVE value of professional knowledge and skill and shared information did not reach the recommended standard of 0.5; however, in the case of the other five factors, it was more than 0.5. Each factor had convergent validity, with a CR range of 0.70–0.96. Discriminant validity is an indicator of whether there is a low correlation among factors for measuring different concepts and is secured by the AVE of a latent variable larger than the square of the correlation coefficient with other factors. Each of the seven factors in this study was found to have discriminant validity (Table 3, Figure 1).

### 4.4. Reliability

Items with at least 0.30 for a correlation with the total score were included, and no items were excluded. The internal consistency reliability coefficient (Cronbach’s α) for the scale was 0.89. Cronbach’s alpha for the subscales were as follows: 0.88, 0.54, 0.69, 0.67, 0.54, 0.56, and 0.90 for Reciprocity, Professional knowledge and skill, Sensitivity, Collaboration, Communication, Shared Information, and Cautiousness, respectively (Table 4).

### 4.5. Criterion Validity

Concurrent validity was evaluated by examining the relationships between the PNPPS-PN and measures of compassion competence (the CCS) and nursing professional values (the NPVS). The PNPPS-PN scale in its entirety and all seven subscales were significantly positively correlated with the CCS and NPVS (Table 4). Pearson’s correlation coefficients ranged from 0.24 to 0.66 (all *p* < 0.001; Table 4).

## 5. Discussion

As more focus is placed on the partnership between medical staff and families in the field of pediatric nursing practice, pediatric nurse–parent partnerships are increasingly accepted as an important concept. The PNPPS-PN is used to measure the perceived partnership between pediatric nurses and parents by pediatric nurses and parents, respectively. Many studies in Korea have used the PNPPS-PN to measure the concept of partnership among pediatric nurses. In contrast, there are no studies that tested the reliability and validity of the PNPPS-PN with only pediatric nurses. This study was conducted to evaluate the appropriateness of the pediatric nurse–parent partnership measurement tool as perceived by pediatric nurses—an important concept in nursing research and practice. After the factor structure of a scale has been identified, it is appropriate to perform CFA [32]. In this study, an attempt was made to adapt and validate the PNPPS-PN developed by Choi and Bang [1] to measure the subjective experience of the pediatric nurse’s nurse–parent partnerships. CFA was performed to reaffirm that the factor structure is appropriate for measuring perceived partnerships between pediatric nurses and parents among pediatric nurses. Through CFA, we validated a 19-item scale consisting of seven factors, the PNPPS-PN. Each of the 19 items is responded to using a 5-point Likert scale, and there are no reverse-scored items. The overall CFA results supported the evidence from the seven factors of the PNPPS-PN when evaluated using a sample of Korean pediatric nurses as satisfactory. The convergent and discriminant validity scores were satisfactory, and the internal consistency reliability was acceptable. In addition, a correlation between each of the seven-factor total scores and each of the total scores of NPVS and CCS was identified. However, the partnership between pediatric nurses and parents may be affected by each country’s healthcare delivery system and culture, thus it is necessary to check the construct validity of pediatric nurses in various countries and settings in the future. If future studies are conducted using PNPPS-PN whose reliability and validity have been verified through this study, it has the advantage of being able to compare and analyze the results of each country’s studies conducted using PNPPS-PN. It is thought that this will provide important basic data for pediatric nurses to objectively understand the perceived partnership with parents and to prepare an appropriate intervention program.

The original scale consisted of seven factors and 34 items; however, in this study, five items with an SMC of 0.5 or less were deleted. A total of five items were deleted in consideration of the factor structure of the original scale. The first deleted question was ‘I immediately help parents who ask for help during the care of their children’. The four items deleted in step 2 included one item relating to the Sensitivity aspect (I encourage those parents who face difficult circumstances.) and three items relating to the Professional knowledge and skill aspect (‘The parents and I have common goals for the care of their children,’ ‘Parents consult with me about problems faced by their children,’ and ‘Sometimes, I am someone parents can rely on’). It is judged that the items were deleted because they were similar to the items in the ‘Collaboration’ aspect. Based on the literature review and the theoretical background of the original scale, it is presumed that these results were obtained because the subjects of this study viewed the partnership as a more equal relationship [14]. Interestingly, the findings supported the original seven-aspect (Reciprocity, Professional knowledge and skill, Sensitivity, Collaboration, Communication, Shared Information, and Cautiousness) scale developed by Choi and Bang [1].

The AVE values checked to verify the convergent validity of the PNPPS-PN in this study reached the recommended standard of 0.5 [33]. Therefore, it can be judged that the convergent validity of the PNPPS-PN in this study was secured.

The CCS and the NPVS were used to test the PNPPS-PN for criterion validity; significant positive correlations were found with the scales, supporting concurrent validity. Nurses’ compassion competence refers to the personal skills and abilities necessary to relieve the difficulties of the patient based on their understanding of the patient [31]. Nursing professionalism is a view of occupational consciousness concerning nursing activities and the job itself, as well as the systematized view of nursing as a profession [35]. This study found that hospital pediatric nurses who perceived that they had positive nurse–parent partnerships were more likely to also have attributes of compassion competence and endorse professional nursing values. Recent studies have confirmed the positive effects of the application of the pediatric nurse–mother partnership program to pediatric cardiac intensive care units [7] and the needs of mothers for such programs [9]. Further research should be conducted to develop an intervention to improve pediatric nurse–parent partnerships and identify additional factors affecting the partnership.

This study has several limitations. First, the PNPPS-PN was validated based on a sample of pediatric nurses at four hospitals in South Korea. Therefore, the results of the analyses may not be generalizable to other pediatric nurse populations. Since this study evaluated the validity of the tool only for child nurses in South Korea, differences with the results for pediatric nurses in other countries cannot be excluded. In other words, differences in the nursing environment such as the scope of work of pediatric nurses in South Korea, a country where scales were developed, and the nursing delivery system, may affect the results. Future research needs to examine the cross-cultural validity of the PNPPS-PN with different populations internationally. Second, the PNPPS-PN was tested in the context of a child’s hospitalization and has limitations in its application to diverse areas of pediatric nursing practice. Therefore, it is necessary to evaluate the validity of the scale in more diverse settings of pediatric nursing practice. Third, the PNPPS-PN was only validated for use with pediatric nurses. Further research should be conducted to validate an instrument for measuring parent–pediatric nurse partnerships for use with parents and compare the degree of partnership perception between pediatric nurses and parents. Finally, some indices of model fit met the criteria less. In the future, it is suggested to develop a theoretical framework that reflects recent trends, and to conduct research that confirms the factor structure and continuously validates it.

## 6. Conclusions

The PNPPS-PN is a psychometrically sound instrument for assessing pediatric nurse-reported partnerships between pediatric nurses and parents in hospital settings. The PNPPS-PN has utility for measuring the strength of pediatric nurse–parent partnerships and can provide data that can be used in educating nurses, research on nursing care, intervention programs, and policies to improve the pediatric nurse and parent partnership. The PNPPS-PN contains seven subscales and 19 items: Reciprocity has five items, Professional knowledge and skill has two items, Sensitivity has four items, Collaboration has two items, Communication has two items, Shared Information has two items, and Cautiousness has two items. Responses to the questionnaire are measured on a 5-point Likert scale, and total scores range from 19–95. The higher the score, the greater the degree of nurse–parent partnership perceived by the pediatric nurse. The instrument has been shown to be valid and reliable and is expected to be useful in understanding partnerships with the parents, as perceived by pediatric nurses, and in accumulating nursing knowledge and developing interventions. Based on these results, the following suggestions can be made for future research. First, the instrument can be used in repetitive research to assess and validate its stability. Second, the instrument can be used to identify the variables affecting pediatric nurse–parent partnerships. Third, the instrument can be used to develop an intervention program to improve pediatric nurse–parent partnerships and determine its effectiveness. Fourth, it is necessary to develop an instrument that can measure partnerships with pediatric nurses from the perspective of the parents.

## Figures and Tables

**Figure 1 healthcare-10-00991-f001:**
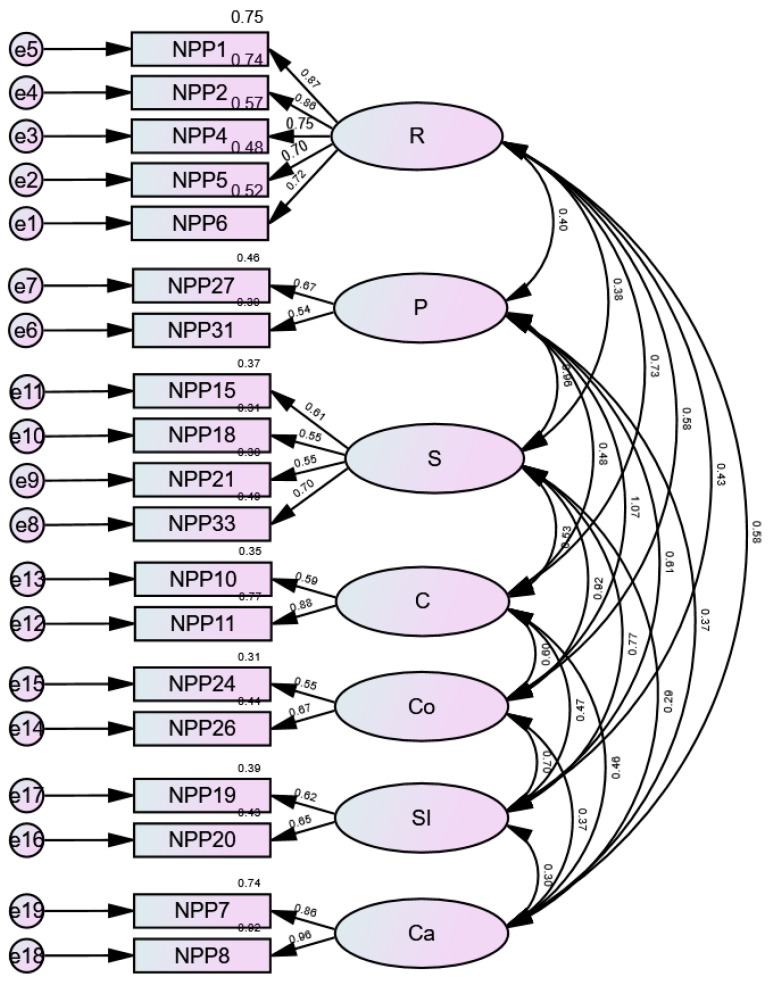
The measurement model for the Pediatric Nurse Parent Partnership Scale-Pediatric Nurse. R: Reciprocity; P: Professional Knowledge and skill; S: Sensitivity; C: Collaboration; Co: Communication; SI: Shared information; Ca: Cautiousness.

**Table 1 healthcare-10-00991-t001:** Characteristics of Participants (*n* = 236).

Characteristics	Categories	*n* (%)	M ± SD	Range
Age (year)	≤29	152 (64.4)	29.29 ± 5.49	23–49
	30–39	63 (26.7)		
	≥40	49 (14.5)		
Marital status	Married	59 (24.9)		
	Not married	178 (75.1)		
Education	Diploma	25 (10.5)		
	Bachelor	168 (70.9)		
	≥Master	44 (18.6)		
Religion	Yes	104 (44.1)		
	No	132 (55.9)		
Current work unit	Pediatric ward	76 (32.1)		
	Pediatric oncology ward	21 (8.9)		
	Newborn and neonatal intensive care unit	100 (42.2)		
	Pediatric intensive care unit	40 (16.9)		
Years of total career			6.53 ± 5.47	1–26
Work duration in pediatric unit			5.90 ± 4.62	1–25.1

**Table 2 healthcare-10-00991-t002:** The Model Fit Indices for the PNPPS-PN.

	χ^2^ (*p*)	df	CMIN/DF	CFI	RMSEA (90% CI)	GFI	NFI	TLI	AIC
Model 1	1180.55 (<0.001)	506	2.33	0.80	0.075 (0.070–0.081)	0.76	0.70	0.78	1358.55
Model 2	876.42 (<0.001)	356	2.46	0.83	0.079 (0.072–0.085)	0.78	0.74	0.80	1034.42
Model 3	309.52 (<0.001)	131	2.36	0.91	0.078 (0.065–0.087)	0.88	0.85	0.88	427.52

Model 1: 7-factor (34 items, original model), Model 2: 7-factor (29 items, modified model), Model 3: 7-factor (19 items, final model), χ^2^ = Chi square; df= degree of freedom; CMIN/DF = Chi square minimum/degree of freedom; CFI = Comparative fit index; RMSEA = Root mean square error of approximation; GFI = Goodness of fit index; NFI = Normed Fit Index; IFI = Incremental Fit Index; TLI = Tucker–Lewis index; AIC = Akaike Information Criterion.

**Table 3 healthcare-10-00991-t003:** Results of the Final Model of the CFA.

Factors	Item	Standardized Estimate (β)	SE	C.R.	R							AVE	CR
					1	2	3	4	5	6	7		
Reciprocity	1	1.00	-	-	1							0.81	0.96
	2	1.00	0.06	16.92									
	4	0.86	0.06	13.65									
	5	0.86	0.07	12.16									
	6	1.01	0.08	12.83									
Professional knowledge and skill	27	1.00	-	-	0.28 **	1						0.63	0.77
	31	0.76	0.10	7.39									
Sensitivity	15	1.01	0.12	8.22	0.32 **	0.60 **	1					0.60	0.86
	18	0.78	0.10	7.49									
	21	0.73	0.10	7.48									
	33	1.00	-	-									
Collaboration	10	0.82	0.11	7.67	0.54 **	0.30 **	0.43 **	1				0.72	0.83
	11	1.00	-	-									
Communication	24	0.79	0.10	7.60	0.43 **	56 **	0.57 **	0.43 **	1			0.58	0.73
	26	1.00	-	-									
Shared information	19	0.71	0.11	6.29	0.33 **	0.34 **	0.50 **	0.35 **	0.42 **	1		0.54	0.70
	20	1.00	-	-									
Cautiousness	7	0.97	0.07	13.21	0.53 **	0.27 **	0.25 **	0.37 **	0.26 **	0.22 **	1	0.92	0.96
	8	1.00	-	-									

SE = Standard error; C.R. = Critical ratio; AVE = Average variance extracted; CR = Construct reliability, ** *p* < 0.01.

**Table 4 healthcare-10-00991-t004:** Correlation Between the PNPPS-PN, Compassion Competence, and Nursing Professional Value/Reliability of the PNPPS-PN.

PNPPS-PN	CCS	NPVS
Reciprocity	0.39 **	0.39 **
Professional knowledge and skill	0.60 **	0.45 **
Sensitivity	0.62 **	0.51 **
Collaboration	0.42 **	0.36 **
Communication	0.55 **	0.48 **
Shared information	0.37 **	0.34 **
Cautiousness	0.35 **	0.24 **
Total	0.66 **	0.57 **

Compassion Competence Scale (CCS); Nursing Professional Value Scale (NPVS). ** *p* < 0.01.

## Data Availability

The data presented in this study are available on request from the corresponding author. The data are not publicly available due to restrictions (e.g., privacy or ethical).

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
