# Peer review of "Factorial Validity of the Pediatric Nurse Parent Partnership Scale—Pediatric Nurses (PNPPS-PN)"

_healthcare, 2022, doi:10.3390/healthcare10060991_

Round 1

Reviewer 1 Report

First of all, congratulations on your work. Please address the recommendations below:

Introduction

Overall, the background allows for an adequate understanding of the research question.

Line 45 includes the concept of "scale" that appears to be inserted without an adequate context. I would suggest rephrasing, focusing on the need for an instrument (e.g. "...little research has been conducted to measure this partnership relationship quantitatively").

As the issue is to validate the scale in question, I recommend describing the concept of partnership before addressing it. To operationalize the recommendation, the concept of "measurement" in lines 49-58 should be the final concept to address (before the last paragraph).

Objective

The objective is clear and in line with the background reasoning.

Methods

The sample size is adequate.

Please indicate the reasoning for the use of the NVPS and CCS. Although understandable, there is no previous mention of the instruments and the concepts in the analysis (only referred to on lines 179-180).

Data analysis is well described but could benefit from a reference to an article or book regarding the criteria for the Goodness-of-fit indices.

Is there a reason for not using modification indices and establishing covariances between errors, and/or removing the outliers through analysis of the observations farthest from the centroid (D2) before analysing models with items removed according to a regression weight <=.50?

The options regarding the discriminant and convergent validity are adequate.

Regarding reliability, could you indicate if there are any items that affect negatively the Cronbach's alpha (SPSS->alpha if item removed)? Other measures could be used (e.g. McDonald's Omega).

Results could have an image of the final model.

Discussion

The discussion is well built.

Please include bibliographic references to the values used as references (e.g. AVE over .5.

Please provide clearer reasoning for indicating the cultural aspect mentioned on lines 300-301.

Conclusion

Try to find a key message to end the conclusion.

Once again, very interesting article.

Author Response

We appreciate the time and effort that you and the reviewers have put into the valuable feedback and insightful comments provided on this manuscript. We have carefully considered each comment and made changes to the manuscript, as required. We have marked the revisions made to the manuscript in red font.

Reviewer 2 Report

Dear authors, 

Thank you for the opportunity to review the manuscript. The manuscript can be of clinical importance. Please find my comments below. The manuscript needs to undergo a substantial revision.

  1. It is not clear in the abstract what does -PN in PNPPS-PN stands for?
  2.  Page 1, Line 27. 'These types of interactions' It is not clear what type of interactions the author wants to elaborate on.
  3.  Paragraph 1 (Introduction). It will be a good idea to reflect on some of the associations between pediatric nurse-parent interaction and outcomes of the pediatric patients.
  4. Lines 52-54. It is evident that PNPPS-PN scale has been used to determine correlations between various variables. In a subsequent sentence, 57-58 authors note that there have not been sufficient studies confirming construct validities... This reflects that there are studies conducted to determine the construct validity of the scale. The authors should give a clear picture of why they are conducting this research if construct validity is already determined. They need to illustrate the methodological issues of such studies.
  5. The introduction focuses more on developing parent-pediatric nurse relationships rather than focusing on the key research question; the validity of the scale. The introduction needs to be restructured and the irrelevant discussion removed (for example lines 82-84). The authors should focus on the lack of construct validity and the potential impact that this could have on the validity of the studies conducted using this scale.
  6. Authors should give a discussion on why they chose the current method (confirmatory factor analysis - just named on the last line (#104) to undertake this study and compare and contrast with potential alternative methods that can be utilized for this study. In other words, there is no clear picture of the rationale of the study design in the introduction.
  7. There is no clear justification of how the participants were selected. Although it is mentioned that the inclusion criteria was 12 months of service, there is no justification of how and why those four hospitals were selected.
  8. The sampling procedure is not provided. 
  9. The justification for sample size needs to be elaborated (lines 113-117). It is not sufficient to tell from one reference that the generally accepted sample size is between 200-400. Besides, the reference that the authors provide for this justification is from a book on Multivariate data analysis. Please provide the page number from which the reference is extracted.
  10. Role of marital status and religion in data collection
  11. The authors have reported the results of Chronbach's alpha in the methods section (Line 140-141, 150, 158). It should be removed from the methods section and reported in the results.
  12. The authors need to report on the validity of the second scale they used for the study (NPVS).
  13. The authors did mention about Chronbach's alpha at many places. However, the authors need to explain what is the normal acceptable value of this indicator and how we need to interpret the value. 
  14. Abbreviations like SPSS and AMOS are not elaborated. Authors need to explain the abbreviations the first time they use them in the article
  15. Line 163...'Factor analysis for scale validation is based on prior theories'. Need to elaborate on which prior theories??
  16. Line 171... A CMIN/DF of 3 or less is acceptable, and 2 or less is good. It means that a value of 2 or less is both good and acceptable. Please be clear and avoid double meanings.
  17. Same comment as above for line 173. It should be rephrased as '0.08 to 0.05 is good' rather than '0.08 or less is good'.
  18. Please check and confirm the ethical approval number (confirm there is no formatting error)
  19. It is not clear how informed consent was obtained. Besides, the authors need to explain how nurses were approached, how many were approached, what was the response rate and how data collection was collected. The methodological details need to be provided. 
  20. Table 1. Can you please illustrate what you mean by religion 'Yes' and 'No'. It is a unique coding for religion and needs to be elaborated
  21. Current work unit - Did the authors collect information on pediatric emergency, pediatric medicine or pediatric surgery?
  22. Table 1 Years of pediatric unit career.... 1-25.1 (please check and confirm). Need to rephrase the title 'Years of pediatric unit career' could be replaced by 'work duration in pediatric unit'
  23. The authors need to illustrate how the three models were determined (lines 196)
  24. Line 198 - Model 2 is the result of removing 1 factor. However, in line 203 and in the explanation of table 2, the authors have mentioned that model 2 contained 29 items. Which one is correct? please clarify
  25. Line 211... Should not we read CMIN/DF value less than 3 as acceptable (line 171) rather than satisfied?
  26. There is no report on the scores obtained from the different scales used in the study. It is important to describe the results.
  27. Line 27 - Discriminant validity - how was it measured - what was the figure? There is no such information in table 3 or in the text. Please illustrate it
  28. Table 4 is not clear. Chronbach's alpha should be removed from the table as it is already described in the text (line 238). It makes readers more confusing about how Chronbach's alpha is calculated from the data displayed in the table.
  29. The authors have calculated different types of validities but have not explained in the introduction why these validities are important to compute. In place of the description of the nurse-parent partnership, the authors should focus the introduction on why it is important to determine these validities and the gap in the literature that these validities for the proposed instrument are previously not determined.
  30. Line 257... 'There are no studies that tested the reliability and validity of the PNPPS-PN with only pediatric nurses'. It does reflect that the tool has been validated with a different group of population. So, where in the literature do the findings from the current study stand with those from previous studies?
  31. Line 264. It is apparent that the authors develop a new scale - PNPPS-PN. Please check if the abbreviation is the same for both scales?
  32. Lines 271-276. What were the criteria to delete the items included in the original scale?
  33. The discussion should also focus on the application of the findings from the study. Whether or not findings from the study will provide better confidence in using the tools.

Author Response

(The authors gave the same response as above.)
